Sponge distribution and the presence of photosymbionts in Moorea, French Polynesia

Freeman Christopher J. 1 2 freemanc@si.edu
Easson Cole G. 3 4
1 Smithsonian Marine Station , Fort Pierce, FL , USA
2 IRCP, Institute for Pacific Coral Reefs, Labex Corail , Papetoai, Moorea , French Polynesia
3 Halmos College of Natural Sciences and Oceanography, Nova Southeastern University , Dania Beach, FL , United States
4 Department of Biology, University of Alabama at Birmingham , Birmingham, AL , USA
Pawlik Joseph
Electronic publication date: 2016 Mar 17
Publication date: 2016
Volume: 4
Electronic Location ID: e1816
Received 2015 Oct 23; Accepted 2016 Feb 24
Copyright: ©2016 Freeman and Easson
Copyright year: 2016
Copyright holder: Freeman and Easson
License: This is an open access article distributed under the terms of the Creative Commons Attribution License, which permits unrestricted use, distribution, reproduction and adaptation in any medium and for any purpose provided that it is properly attributed. For attribution, the original author(s), title, publication source (PeerJ) and either DOI or URL of the article must be cited.
License URL: https://creativecommons.org/licenses/by/4.0/

Keywords: Symbiosis, Photosymbionts, Porifera, Biodiversity, Moorea, Mutualism

Funding: Institute for Pacific Coral Reefs (IRCP) Smithsonian MarineGEO Tennenbaum Marine Observatories Network This work was supported by a research grant awarded to CJF from the Institute for Pacific Coral Reefs (IRCP), as well as a postdoctoral fellowship awarded to CJF from the Smithsonian MarineGEO and Tennenbaum Marine Observatories Network. The funders had no role in study design, data collection and analysis, decision to publish, or preparation of the manuscript.

==============================
Photosymbionts play an important role in the ecology and evolution of diverse host species within the marine environment. Although sponge-photosymbiont interactions have been well described from geographically disparate sites worldwide, our understanding of these interactions from shallow water systems within French Polynesia is limited. We surveyed diverse habitats around the north coast of Moorea, French Polynesia and screened sponges for the presence of photosymbionts. Overall sponge abundance and diversity were low, with <1% cover and only eight putative species identified by 28S barcoding from surveys at 21 sites. Of these eight species, seven were found predominately in shaded or semi-cryptic habitats under overhangs or within caverns. Lendenfeldia chondrodes was the only species that supported a high abundance of photosymbionts and was also the only species found in exposed, illuminated habitats. Interestingly, L. chondrodes was found at three distinct sites, with a massive, fan-shaped growth form at two of the lagoon sites and a thin, encrusting growth form within a bay site. These two growth forms differed in their photosymbiont abundance, with massive individuals of L. chondrodes having higher photosymbiont abundance than encrusting individuals from the bay. We present evidence that some sponges from French Polynesia support abundant photosymbiont communities and provide initial support for the role of these communities in host ecology.

Introduction

In the marine environment, symbiotic interactions fuel high species diversity in otherwise nutrient poor systems like hydrothermal vents and coral reefs (Steinert, Hentschel & Hacker, 2000; Venn, Loram & Douglas, 2008). On tropical reefs, the mutualism between the dinoflagellate symbiont Symbiodinium and reef building corals has long epitomized such an interaction (Muscatine & Cernichiari, 1969; Muscatine & Porter, 1977), but recent research has explored the functional role of photosynthetic microbes within other dominant benthic fauna like octocorals (Baker et al., 2015) and sponges (Freeman et al., 2013). Although autotrophic nutrition supplied by these photosymbionts can provide more than enough C to compensate for holobiont (microbial symbiont and host) metabolism (Venn, Loram & Douglas, 2008; Usher, 2008), there is substantial variation in the structure and function of these interactions across groups and species (Thacker & Freeman, 2012).

While reef building corals and octocorals host diverse communities of metabolically distinct microbes (Knowlton & Rohwer, 2003), many species of sponges support a microbial diversity that rivals that found in other eukaryotic hosts (Thacker & Freeman, 2012). The structure of these interactions also varies across species, with high host specificity in microbial community composition across sympatric host species (Thacker & Freeman, 2012; Easson & Thacker, 2014). This, coupled with the fact that host sponge identity accounts for over 70% of the variation in the placement of an individual sponge within bivariate (δ13C and δ15N) isotopic “niche space,” implies that microbial community composition also plays an important role in the functional placement of a species in its ecosystem (Freeman, Easson & Baker, 2014).

By allowing sponge hosts to utilize inorganic sources of C and N, abundant photosymbiont communities also play a crucial role in sponge metabolism and impact the overall nutritional benefit of these interactions to the host (Freeman & Thacker, 2011; Freeman et al., 2013; Freeman, Easson & Baker, 2014). Although sponge-photosymbiont interactions have been widely reported in disparate geographic areas ranging from the Great Barrier Reef (Wilkinson, 1983), Palau and Guam (Ridley, Faulkner & Haygood, 2005; Usher, 2008), Western Australia (Lemloh et al., 2009), Africa (Steindler, Beer & Ilan, 2002), the Red Sea and Mediterranean (Wilkinson & Fay, 1979; Usher, 2008), and the Caribbean (Erwin & Thacker, 2007; Freeman & Thacker, 2011; Easson & Thacker, 2014), limited data exist on the prevalence of these interactions in other locations. For instance, despite the fact that over 75 species of sponges have been previously reported in the Society and Marquesas Islands of French Polynesia, the presence of photosymbiont hosting species is largely based on anecdotal evidence that many of the species catalogued are from orders dominated by “phototrophic” species (Hall et al., 2013). In addition, such assertions may not be accurate, as the presence of photosymbionts can be variable even among species within a genus and may thus vary substantially at higher taxonomic levels (Erwin & Thacker, 2007).

To identify sponge species hosting photosymbiont communities in Moorea, French Polynesia and describe the general habitat in which they are found, we conducted surveys across diverse habitats and environmental conditions in bays, lagoons, and along the reef slope on the northern coast of Moorea. Surveys included both man-made and natural substrates, as well as under overhangs and semi-cryptic habitats. Sponges were identified by obtaining a partial 28S rRNA gene sequence (Thacker et al., 2013; Erpenbeck et al., 2015) and screened for the abundance of photosymbionts by chlorophyll a analysis (Erwin & Thacker, 2007).

Figure 1 Map of sites surveyed along the northern coast of Moorea, French Polynesia.

Sponges were observed at sites represented by a red marker, while white markers denote sites in which sponges were not observed. Abbreviations on red markers correspond to site names in Table 1. Map data: Google, Digital Globe Image NASA and CNES/Astrium.

Materials and Methods

Sample collection

We surveyed 21 sites along the northern coast of Moorea, French Polynesia (see Schrimm, Buscail & Adjeroud, 2004 for a detailed description of the island) using a combination of SCUBA and snorkeling (Fig. 1). Sites included locations previously reported to support sponge communities (Adjeroud & Salvat, 1996; Adjeroud, 1997; Desmet, 2009; Hall et al., 2013) and also sites that were selected haphazardly by satellite images and word of mouth from local researchers. To include both diverse habitats and substrate types, we surveyed bay and lagoon sites, including the motu (an independent set of islands within the lagoon), reef crest, channel openings, near beaches, reef slopes, and docks/man-made structures (Fig. 1; Table 1). Initial transects revealed that average sponge cover was <1%, with coral or bare rock as the most common substrate. To adequately sample sponge species in this area, we therefore adapted our methods and surveyed sites by a two-diver swimming census method. Using this method, when a sponge was observed, a small (3–6 cm3) sample was collected using a sharp dive knife and placed into an individual sealed bag. Once on the boat, samples were stored on ice for transit back to the Centre de Recherches Insulaires et Observatoire de L’Environment (CRIOBE) research station. At CRIOBE, samples were photographed and two subsamples were taken. The first subsample was preserved in 95% EtOH for taxonomic identification via 28S barcoding. The use of 28S barcoding for sponge molecular taxonomy has increased in recent years (Thacker et al., 2013; Erpenbeck et al., 2015), and while species-level taxonomic resolution may be limited using this marker, it is not uncommon for sponges of French Polynesia to be identified to only genus via morphological characters alone (Adjeroud & Salvat, 1996; Adjeroud, 1997; Desmet, 2009; Hall et al., 2013). We therefore propose that 28S barcoding provides comparable taxonomic resolution to morphology-based studies in this area and, importantly, also provides a comparatively rapid, effective, and objective comparison of specimens collected across sites that may have slightly different morphologies. Because of this, we have intentionally restricted our identifications to 28S barcoding. The second subsample was frozen for chlorophyll a analysis. Permits for sponge collection and export were obtained from the Institute for Pacific Coral Reefs (IRCP) and the Haut-commissariat de la République en Polynésia Francaise. Representative vouchers of these sponges are stored at the Smithsonian Marine Station in Fort Pierce, Florida.

Table 1 Sites supporting sponge communities around Moorea, French Polynesia.

Site abbreviations correspond to labeled red markers in Fig. 1.

Site	Name	GPS coordinates	Description	Sponge species	
S&G	Spur and Groove	17°30′7.21″S149°55′36.98″W	6–8 m deep, hardbottom, high-energy	Verongida sp. and Dictyoceratida sp.	
MR	Motu Reef	17°29′16.50″S149°55′7.73″W	1–2 m deep, coral heads, hardbottom, sandy habitat	Lendenfeldia chondrodes (De Laubenfels, 1954) massive fan morph	
MC	Motu Channel	17°29′20.37″S149°54′52.47″W	2–4 m deep, sandy bottom, occasional coral head and hard substrate, high current	Lendenfeldia chondrodes (De Laubenfels, 1954) massive fan morph	
ICR	Intercontinental Reef	17°29′17.21″S149°53′47.82″W	2–3 m deep, vertical wall in channel, hard substrate, moderate current	Lendenfeldia chondrodes (De Laubenfels, 1954) massive fan morph	
ICL	Intercontinental Lagoon Wall	17°29′25.97″S149°53′32.56″W	1 m deep, artificial concrete substrate	Haliclona sp.	
OB3	Opunohu Bay Site 3	17°29′48.71″S149°51′46.03″W	2–10 m deep, hard substrate, overhangs and high vertical relief, moderate current	Lendenfeldia chondrodes (De Laubenfels, 1954) encrusting, smooth morph, Leucetta sp., Heteroscleromorpha sp, Cinachyrella sp.	
OB2	Opunohu Bay Site 2	17°29′53.60″S149°51′42.36″W	2–10 m deep, hard substrate, overhangs and high vertical relief, moderate current	Lendenfeldia chondrodes (De Laubenfels, 1954) encrusting, smooth morph, Cinachyrella sp.	
OB1	Opunohu Bay Site 1	17°30′19.68″S149°51′29.17″W	3–5 m deep, hard substrate, overhangs	Lendenfeldia chondrodes (De Laubenfels, 1954) encrusting, smooth morph, Heteroscleromorpha sp., Cinachyrella sp., Leucetta sp.	
CB	Cooks Bay	17°29′32.09″S149°49′34.26″W	2–5 m deep, hard substrate, overhangs	Lendenfeldia chondrodes (De Laubenfels, 1954) encrusting, smooth morph, Leucetta sp., Heteroscleromorpha sp.	
KD	Kaveka Hotel Dock	17°29′23.59″S149°49′8.37″W	1–2 m deep, artificial concrete substrate	Haliclona sp., Leucetta sp., Dysidea sp.	

DNA extraction, PCR amplification, and sequencing

A total of 40 collected specimens were sequenced to obtain a molecular barcode. Genomic DNA was extracted from ethanol-preserved samples using the PowerSoil DNA extraction kit (MoBio, Carlsbad, CA, USA), following the manufacturer’s protocol. The oligonucleotide primers SP635F and SP1411R were used to amplify a portion of the 28S ribosomal subunit (Thacker et al., 2013), yielding an approximately 650bp fragment. PCR reaction products were gel-purified and cleaned using the Wizard PCR Preps DNA Purification System (Promega, Madison, WI, USA). Forward and reverse sequencing reactions were performed at the University of Alabama at Birmingham (UAB) Center for AIDS Research (CFAR) DNA Sequencing Core Facility. Forward and reverse sequences were then compared to ensure the accuracy of sequencing reactions in CodonCode Aligner software (CodonCode, Dedham, MA, USA) and Geneious (version 6.1.8; Biomatters Limited, Auckland, New Zealand), yielding a final consensus sequence for each voucher specimen. A representative sequence for each species is archived in GenBank under accession numbers KU746954, KU746955, KU746956, KU746957, KU746958, KU746959, KU746960 and KU746961.

Phylogenetic analyses and species identification

Phylogenetic relatedness among surveyed sponge species was assessed using a phylogeny constructed from a partitioned alignment of gene sequences from 40 collected specimens and 44 GenBank sequences coding for the large (28S) nuclear ribosomal subunits, which is a common marker used for molecular identification of sponge species (Thacker et al., 2013). Sequences were aligned using the default options of MAFFT 7.245 (Katoh et al., 2002). We implemented a relaxed-clock model in MrBayes version 3.2.1 (Ronquist et al., 2012), using the CIPRES computational resources (Miller, Pfeiffer & Schwartz, 2010) and constrained sponges in the class Calcarea as an outgroup using the independent gamma rate relaxed clock model with a birth-death process (Aris-Brosou & Yang, 2003). We included three parallel runs of 10 million generations, each using four Markov chains and sampling every 100 generations. A consensus phylogeny of the three parallel runs was summarized following a burn-in of 25%. Sponge specimens were ultimately identified to the lowest possible taxonomic level using the 28S genetic marker.

Chlorophyll a analysis

At the Smithsonian Marine Station, frozen sponge samples were lyophilized overnight, ground to a fine powder, and weighed to the nearest 0.001 g. To quantify photosymbiont abundance, chlorophyll a (chl a) analysis was carried out as in Erwin & Thacker (2007) and Freeman, Easson & Baker (2014). In short, dried sponge tissue was extracted in 90% acetone overnight at 4 °C and the concentration of chl a was quantified by measuring the absorbance of these extracts at 750, 664, 647, and 630 nm and applying equations from Parsons, Maita & Lalli (1984). Chlorophyll a values were not obtained for Dysidea sp. from Kaveka dock because these sponge individuals were too small to allow for both taxonomic vouchers and chl a samples to be taken. We compared differences in mean chl a concentrations using an Analysis of Variance (ANOVA; Systat, v. 11). Pairwise comparisons of mean chl a concentrations between the three L. chondrodes populations and between L. chondrodes and the low chl a samples were carried out using the Fisher’s least significant difference (LSD) post hoc test.

Figure 2 Photographs of sponges from Moorea, French Polynesia: Heteroscleromorpha sp. from Opunohu Bay site #1 (A); Cinachyrella sp. from Opunohu Bay site #2 (B); Haliclona sp. from Intercontinental Lagoon Wall (C); Lendenfeldia chondrodes from Intercontinental Reef (D), Motu Channel (E and F), and Opunohu Bay Site #3 (G); Dictyoceratida sp. from the Spur and Groove site (H); Verongida sp. from Spur and Groove Site (I–K); Leucetta sp. from Cooks Bay (L). Photographs of Dysidea sp. from Kaveka Hotel Dock were not taken.

Results

Almost all specimens were from the class Demospongiae, with only one species from Calcarea that closely matched Leucetta sp. Within Demospongiae, sequenced specimens spanned all three subclasses (Heteroscleromorpha, Keratosa, Verongimorpha), and five sponge orders within these subclasses (Hadromerida, Spirophorida, Haplosclerida, Verongida, Dictyoceratida). Many of these specimens were genetically similar to members of genera previously reported in French Polynesia. For instance, CB# 1–5 (Cooks Bay, Figs. 1 and 2) were similar to species from the genus Leucetta sp. (Adjeroud & Salvat, 1996; Adjeroud, 1997; Hall et al., 2013; Fig. 3), OB2# 1 (Opunohu Bay 2, Figs. 1 and 2) was similar to Cinachyrella sp. (Hall et al., 2013; Fig. 3), IC# 1 to 3 and KD# 3 and 4 (Intercontinental Lagoon and Kaveka Dock; Figs. 1 and 2), were identified as a Haliclona sp. (Adjeroud, 1997; Hall et al., 2013; Fig. 3), and KD# 1 and 2 (Kaveka dock, Fig. 1) were most similar to a species within the genus Dysidea (Adjeroud & Salvat, 1996; Adjeroud, 1997; Hall et al., 2013; Fig. 3). Other specimens were genetic matches to sponges within the orders Verongida (S&G# 6–9), Dictyoceratida (S&G# 1–5), and the subclass Heteroscleromorpha (OB1#1 and 2) (Figs. 1–3).

Figure 3 Bayesian 28S phylogeny of relationships among collected specimens and closely related, previously sequenced specimens from GenBank.

Scale bar indicates 0.06 substitutions per site. Posterior probabilities are shown for each node.

One species included several individuals of variable growth forms collected from different locations (an encrusting growth form in Opunohu Bay [OB3# 1–6] and a massive fan-shaped growth form in the Motu Channel [MC# 1–5] and Intercontinental Reef [ICR# 1–5]; Figs. 1 and 2) that were shown to all be genetically identical (Fig. 3). BLAST results in GenBank revealed that these samples most closely matched Fasciospongia chondrodes (also known as Lendenfeldia chondrodes). Histological comparisons indicated internal fiber structure and thickness consistent with L. chondrodes (C Diaz, pers. comm., 2015), and 16S and ITS cloning of all three morphotypes (C Freeman & C Easson, 2016, unpublished data) showed the presence of both Oscillatoria spongeliae and Synechocystis sp. symbionts, a composition unique to L. chondrodes (as in L. chondrodes from Ridley, Faulkner & Haygood, 2005). These three separate observations offer compelling evidence that these represent different morphotypes of the same species.

Most sponge species were found in shaded or otherwise semi-cryptic habitats, except for Lendenfeldia chondrodes, which was found in exposed locations at all three sites. Interestingly, there was a wide range of chlorophyll a values across sponge samples (Analysis of Variance [ANOVA]: F = 65.270, p < 0.001; Fig. 4), with six of the seven species having low (<125 µg chl a [g sponge tissue]−1; Erwin & Thacker, 2007) values and all three populations of Lendenfeldia chondrodes having significantly higher (∼300–675 µg chl a [g sponge tissue]−1) chl a values than any of the other species (ANOVA followed by LSD multiple pairwise comparisons: p < 0.05; Fig. 4). Chlorophyll a values of L. chondrodes varied across the three sites (ANOVA followed by LSD multiple pairwise comparisons: p < 0.05), with the encrusting, smooth growth form in Opunohu Bay having the lowest photosymbiont abundance and the two massive, fan-shaped growth forms from the Motu and Intercontinental reef having chl a values over 550 µg chl a (g sponge tissue)−1 (Fig. 4). The highest chl a values were found in the massive, fan-shaped growth form from the Intercontinental reef.

Figure 4 Mean (±) SE chlorophyll a concentration (μg chl a [g sponge tissue]−1) across seven sponge species from habitats around Moorea, French Polynesia.

The dotted line at 125 μg chl a (g sponge tissue)−1 denotes the separation between sponges hosting high (>125) and low (<125) photosymbiont communities (established by Erwin & Thacker, 2007). Sponge identifications and sites are: 1: Leucetta sp. from Kaveka Dock (KD), 2: Leucetta sp. from Cooks Bay (CB), 3: Dictyoceratida sp. from Spur and Groove (S&G), 4: Verongida sp. from Spur and Groove (S&G), 5: Cinachyrella sp. from Opunohu Bay (OB), 6: Heteroscleromorpha sp. from Opunohu Bay (OB), 7: Haliclona sp. from Kaveka Dock (KD), 8: Haliclona sp. from Intercontinental Lagoon (ICL), 9: Lendenfeldia chondrodes from Opunohu Bay (OB), 10: Lendenfeldia chondrodes from Motu Channel (MC) and 11: Lendenfeldia chondrodes from Intercontinental Reef (ICR).

Discussion

This study expands our understanding of sponge distribution in Moorea, French Polynesia and provides initial quantitative evidence that some sponges in French Polynesia host abundant photosymbiont communities. The overall sponge diversity throughout the Society Islands has previously been shown to be around 40 species (identified as operational taxonomic units [OTUs]; Hall et al., 2013), with up to 17 species across sites in Moorea (10 species: Adjeroud & Salvat, 1996; 17 species: Adjeroud, 1997; 7 species: Desmet, 2009; 12 species: Hall et al., 2013). Variation in overall sponge diversity reported from Moorea likely reflects the fact that some studies survey deep sites (18–39 m; Hall et al., 2013), while others restrict collections to bays (Adjeroud & Salvat, 1996) or survey a wide variety of habitats and depths (Adjeroud, 1997). To search for photosymbiont-hosting sponges, our surveys were generally restricted to shallow (<10 m deep) environments, but we sampled across diverse environmental conditions in bays, lagoons, and channel openings (Adjeroud & Salvat, 1996; Adjeroud, 1997; Schrimm, Buscail & Adjeroud, 2004).

Our qualitative estimates of <1% sponge cover on shallow substrates around Moorea approaches the lower range reported from other locations in the Pacific. For instance, the abundance of individual sponge species was from <1% to ∼10% in the Wakatobi region of Indonesia, and the percent cover of the whole sponge community in this region ranged from 10% to over 50%, (Bell & Smith, 2004; Bell et al., 2010), with over 100 sponges per m2 at some sites (Powell et al., 2014). Likewise, the total number of individual sponges observed across a depth gradient from 5 to 40 m on Davies Reef on the Great Barrier Reef (GBR) ranged from 25 to well over 2,000 at a single site, with over ten individuals per m2 at some sites (Wilkinson & Evans, 1989) and between 0.5 to ten sponge individuals per m2 at sites across the entire GBR reef tract (Wilkinson, 1987). Sponge abundance in Moorea is also substantially lower than that observed from most sites in the Caribbean, where the percent cover of sponges above 15 m depth ranges from 0.6% to over 20% (Pawlik et al., 2015).

We found interesting trends in the distribution of sponges across sites. Species with low photosymbiont abundance were found in semi-cryptic habitats underneath overhangs and in caverns. In addition, the Haliclona sp. was found in high abundances on artificial substrates surrounding the Intercontinental Hotel and on the pilings of the Hotel Kaveka within Cooks Bay, and rarely on reefs. Whether this species preferentially settles on artificial substrates or favors areas with potentially elevated nutrient levels remains to be tested. In contrast, the three growth forms of Lendenfeldia chondrodes were all found on exposed substrates in generally well-lit environments. Together this certainly suggests that environmental conditions are playing a role in the distribution of sponges in Moorea.

The distribution of sponges on Pacific reefs is commonly ascribed to environmental conditions. In particular, across the Great Barrier Reef, the overall biomass of sponges decreases as you move offshore and the presence of phototrophic sponges (obtaining at least 50% of their energy requirements from photosynthetic symbionts) increases, with up to 90% of sponges from offshore reefs being phototrophic (Wilkinson, 1987). Although we found three sponges in the order Dictyoceratida (Hall et al., 2013), only Lendenfeldia chondrodes had chl a concentrations that were high enough to suggest that it might be phototrophic.

Unlike phototrophic sponges on the GBR, L. chondrodes was found at relatively pristine lagoon sites, at a more turbid site close to development (Intercontinental Reef), and within bays that are impacted by terrestrial sources of nutrients (Adjeroud & Salvat, 1996; Adjeroud, 1997; Schrimm, Buscail & Adjeroud, 2004). In addition, while phototrophic sponges in the Pacific typically have foliose, flattened plate or cup like growth forms to maximize light exposure (Wilkinson, 1987), individuals of L. chondrodes in Moorea were either thinly encrusting or encrusting with abundant vertical fingers, giving the sponge a fan-shaped growth form (similar to L. chondrodes in Ridley et al., 2005; Fig. 2). These two growth forms appear to be intermediate along the continuum encompassing foliose plate sponges from the GBR, encrusting and finger-like projections in sponges from Palau and Guam, and some of the encrusting and generally amorphous or fan-shaped sponges that host photosymbionts in the Caribbean (Thacker & Starnes, 2003; Ridley et al., 2005; Freeman & Thacker, 2011; Freeman et al., 2013; Pawlik et al., 2015).

Lendenfeldia chondrodes has been previously shown to host photosymbionts, including the sponge-specific filamentous cyanobacteria Oscillatoria spongeliae, as well as Synechocystis sp. (Ridley, Faulkner & Haygood, 2005). This, coupled with our data showing high photosymbiont abundance within individuals of L. chondrodes in Moorea, suggests that this sponge is likely benefitting from this association. Additional work is underway to assess the productivity of this species and determine if their internal photosymbionts are indeed capable of more than compensating for host and symbiont metabolism (Freeman et al., 2013; Pawlik et al., 2015). Regardless of its status as a phototrophic species, L. chondrodes was one of the dominant species across sites, suggesting that the presence of photosymbionts may allow this species to survive and grow across more diverse habitats than species that do not host photosymbionts and are largely restricted to refugia.

It is important to mention that cyanobacteria may acclimate to local conditions by increasing or decreasing the chl a concentrations within each cell (Six et al., 2004). Despite this, we anticipate that it is unlikely that variation in chl a concentration per cell is structuring the vast differences between Lendenfeldia chondrodes and the other species in this study that have chl a values ranging from 1.5% to 28% of the chl a values of L. chondrodes from Opunohu Bay and 0.7 to 13.5% of the chl a values of L. chondrodes from the Intercontinental Reef site. In addition, while cyanobacterial cell counts performed in conjunction with traditional chl a pigment analyses may allow for a more robust estimate of chl a per cell, the difficulty in resolving and counting individual cells within the filaments of Oscillatoria spongeliae via fluorescent microscopy has been demonstrated previously (Flatt et al., 2005). Ongoing research is therefore aimed at determining how symbiont metabolism varies across Lendenfeldia chondrodes collected from these three sites, and whether this variation is correlated with these differences in chl a concentration.

A recent study by Wecker et al. (2015) identified dinoflagellates from the genera Symbiodinium and Dinophysis, as well a member of the order Dinophyceae within three sponge species from Moorea and Tahiti. Wecker et al. (2015) identified Symbiodinium and other dinoflagellates by PCR screening, but without quantifying the abundance of these symbionts via chl a analyses or microscopy, it is difficult to resolve whether their presence reflects a true symbiosis or just reflects a low abundance of these microbes in the sponge at the time of sampling. In contrast, L. chondrodes from our study has been shown by amplification of the 16S rRNA gene and transmission electron microscopy to host abundant communities of sponge-specific cyanobacteria (Ridley, Faulkner & Haygood, 2005). We thus anticipate that elevated chl a values in this species predominately reflect cyanobacterial symbiont abundance and not Symbiodinium or other dinoflagellates.

In summary, photosymbionts may allow sponges like L. chondrodes to compete for space in well-illuminated reef habitats, filling niches that are unavailable to other sponge species (Freeman, Easson & Baker, 2014). In Moorea, these sponges may also expand into niches recently vacated by coral die-offs from crown-of-thorns outbreaks (Kayal et al., 2012). Manipulative shading experiments and reciprocal transplants among these sites may help to elucidate this further (Thacker & Freeman, 2012). With recent evidence that the functional placement of a sponge individual within an ecosystem may be driven by the overall microbial community composition, however, additional work is needed to investigate variation in the microbiomes of sponges from across French Polynesia.

Supplemental Information

Supplemental Information 1 Raw data

Raw chlorophyll a data for sponge species from sites around Moorea, French Polynesia

Click here for additional data file.

We thank the staff and scientists at the CRIOBE research station on Moorea for their incredible logistical support of this project. We also thank Dr. Cristina Diaz for her help in verifying the morphotypes of L chondrodes. This is contribution number 1028 from the Smithsonian Marine Station at Fort Pierce.

Additional Information and Declarations

Competing Interests

Author Contributions

Field Study Permissions

Data Availability

The authors declare there are no competing interests.

Christopher J. Freeman and Cole G. Easson conceived and designed the experiments, performed the experiments, analyzed the data, contributed reagents/materials/analysis tools, wrote the paper, prepared figures and/or tables, reviewed drafts of the paper.

The following information was supplied relating to field study approvals (i.e., approving body and any reference numbers):

Permits for sponge collection and export were obtained and signed by the Institute for Pacific Coral Reefs (IRCP) and the Haut-commissariat de la Republique en Polynesia Francaise (High commission of the republic of French Polynesia).

The following information was supplied regarding data availability:

A representative sequence for each species is archived in GenBank under accession numbers KU746954, KU746955, KU746956, KU746957, KU746958, KU746959, KU746960 and KU746961.

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
