# Peer review of "Sponge distribution and the presence of photosymbionts in Moorea, French Polynesia"

_PeerJ, doi:10.7717/peerj.1816_

## Round 0.1 · original submission · Major Revisions

RE: Freeman & Easson: sponge distribution in Moorea

I now have two reviews of the above-referenced ms., one recommending minor revisions, the other rejection. While I concur with Reviewer 1 that any contribution to our understanding of sponges on Indo-Pacific reefs is needed and valuable, I also agree with both reviewers that this ms. has important deficiencies in its current form. I hope the authors will be able to address the criticisms in a major revision that addresses these topics in particular:

(1) “sponge distribution.” It is quite surprising that, considering the recent interest in comparisons between the communities of sponges on Caribbean and Pacific coral reefs (see last paragraph, Reviewer 1), the authors have chosen to ignore this issue in a paper that purports to address sponge distributions. Further, there are no real data on overall sponge abundance (percentage cover or biomass) that can be used to compare this island with any other tropical reef location. Generally, when “surveys” are conducted, transects or quadrats are used and quantitative data result. Can the authors provide abundance data for their survey sites, even if the data are estimates at the low end (e.g., <1% cover, or similar) because sponges are so scarce that great spans of reef bottom have to be passed over before any are observed? Are there similar data for other sites in the Pacific? I know the Bell group has published data for Wakatobi (2010, Mar Biol), I’m sure there are other data for Indo-Pacific sites, including the GBR (see Wilkinson’s papers). If the sponge cover is very low, what are the dominant organisms at these sites, or is it bare rock? Can the authors provide some comparison to sponge abundance or diversity on Caribbean reefs (see tables in the Loh & Pawlik 2014 PNAS paper)?

(2) Dominance of photosymbiont-containing sponge species. Again, this is an issue of great recent interest in sponge ecology that is not placed in proper context. Beyond the Caribbean/IndoPacific comparison that should be addressed, how does the dominance of L. chondrodes on this island compare to the described distribution of photosymbiont-containing species at other locations in the IndoPacific? How does the growth form of this species compare to other photosymbiont-containing species reported from reefs in the IndoPacific?

(3) Taxonomy. I concur with both reviewers regarding the lack of morphological characteristics of the sponges characterized in this study that compliment the molecular analyses. Can basic morphological descriptions be provided? If the authors have photos and tissue samples, this should not be a difficult addition to this contribution. One of the wonderful things about PeerJ is the potential inclusion of easily accessed supplemental information, including photographs.

Reviewer 1 ·

Basic reporting

Line 39: Add “some” before sponges here, as really only one species exhibited evidence of abundant photosymbiont communities.

Line 108: Typo. Capitalize “institute.”

Line 121: Please add the location and identifiers for voucher specimens.

Lines 127-129: Typo? This sentence is repetitive of the information in the preceding sentence concerning phylogenetic reconstruction and the targeted gene sequences.

Line 154: Typo. Add “the” between “from class.”

Lines 223-224: Table 1 indicates that all samples occurred at 10m or less.

Figure 3: Typo. Do not capitalize “Chlorophyll” here.

Experimental design

Line 106-107: Clarify. No traditional taxonomic identification was reported (gross morphology, spicules, etc.) only DNA barcoding.

Line 137: Please clarify “physical identifiers.” The paper presents no morphological characterization of the sponges collected, making future identifications and comparisons to this work difficult.

Line 168: Again, how were the current sponge samples compared to these vouchers?

Lines 177-178: Again, please define the characteristics used in the “visual match.”

Line 204: Again, same. How where the species collected compared to vouchers in the Moorea Biocode?

Lines 208-212: Species in the families Pseudoceratinidae and Aplysinidae can be distinguished readily based of fiber skeleton branching (dendritic in the former, reticulated in the latter), requiring only basic light microscopy work.

Figure 2: Please add posterior probability support values to each node, otherwise it is difficult to assess the confidence that we can have in various branches of the phylogeny.

Figure 3: Please add a statistical account of these data (e.g. one-way ANOVA with post-hoc pairwise comparisons).

Validity of the findings

Line 160: This sentence highlights the major issue with this work – the absence of morphological taxonomy to accompany their DNA sequence data. This is particularly troublesome here because the authors present only DNA-based phylogenetic data to identify their sponge species, yet make conclusions against their findings based on undescribed comparisons to other sponge vouchers. How was this sample identified as Stylissa massa? Why does this evidence trump the phylogenetic evidence at such a broader (ordinal) level?

Lines 182-185: The phylogeny does not shown any identical matches for these samples. In fact, Figure 2 shows the sequences were closely related to Fasciospongia chondrodes and two species of Lamellodysidea. Was the referenced BLAST sequence not used in phylogenetic construction? Again, the comparison to vouchers is not detailed, thus it is difficult to assess what “closely resemble” means.

Lines 206-207: This again highlights the difficulty with the current work. How can we build on these results when appropriate taxonomic descriptions are not offered for future comparisons? The barcoding is an excellent objective dataset for future efforts, but even basic morphological characterization would very useful for in situ identifications.

Lines 267-268: This sentence overstates the results of the study. As stated in the abstract, the authors find “initial support” for the role of photosymbionts in sponge ecology in Moorea. Stating that these communities play a “critical role” is beyond the reach of the current data, as is the conclusion that these photosymbionts play a critical role in sponge evolution.

Additional comments

This research on sponge diversity and photosymbiont communities in Moorea is a needed addition to the literature and the authors are commended for their DNA barcoding and chlorophyll a analyses.

My main concern is the apparent lack of morphological taxonomy to allow for future identifications and comparisons with the current work. This is particularly troublesome when the authors contradict their phylogenetic results to identify a species (e.g. Sylissa massa) and suggest their inventory includes new species (e.g. lines 177-178). Basic descriptions of the investigated sponges would add significantly to their manuscript and the accessibility of the data therein. Minor concerns are detailed elsewhere.

Finally, I also suggest that the authors couch their findings in the context of the re-emergent controversy on the trophic ecology of sponges (see Pawlik et al. 2015 MEPS 519: 265–283 and subsequent comments/replies). Their work provides further insight into this issue, showing an overall low diversity of sponges in an oligotrophic Pacific reef and non-photosymbiotic species restricted to refugia (lines 188, 228-230). Further, the photosymbiotic sponge (L. chondroides) appears to dominate the community (present in 7 of 10 sites with sponges, Table 1) and exhibited massive growth forms with higher photosymbiont abundance (lines 194-197, 248-250).

Reviewer 2 ·

Basic reporting

I found the title and abstract of the manuscript misleading. The bulk of work in the study is sponge identification and not analysis of host-symbiont interactions.
If the intention of the authors is to publish an article on host-photosymbiont interactions, the data presented here are too preliminary. Considering the future work suggested in the discussion section (lines 243-254) the work seems to have been artificially subdivided. The work presented in this manuscript should be presented with the results of suggested work as part of the methods section (selecting the sponge of interest) and a paragraph with figure in results (chlorophyll a).
Figures- while figure 1 and 2 are relevant to the content of the article they are redundant. All the data in the figures is found in table 1.

Experimental design

The research question, presented in the title, is clearly defined. However there are no statements on how the study contributes to filling the gap. The authors state the aim of their study is to "identify sponge species hosting photosymbiont communities in Moorea, French Polynesia and describe the general habitat in which they are found".
The investigation could have been conducted more rigorously, especially in the case of Lendenfeldia chondrodes. Measuring chlorophyll a concentration is not sufficient to claim difference in photosymbiont abundance. Two symbiotic communities with the same cell abundance can differ in chlorophyll a concentration per cell, ending with the same overall result.
In order to show an effect of photosymbionts on sponge distribution the authors should test for the presence of the symbiotic cyanobacteria (or other photosymbionts) in larva and analyze dispersion. This will show causation and not only correlation. Looking at fully grown specimens, one cannot conclude if the distribution is affected by symbionts or effects symbionts.

Validity of the findings

The data in itself is robust but in several cases, additional examination is warranted:
• A more thorough comparison should be made between morphotypes of L. chondrodes to verify they are of the same species.
• A second independent measurement of photosymbiont abundance is needed.
No statistical test were performed to support the descriptive data.
The data presented in this work cannot support any conclusions regarding sponge distribution and ecology. The article contains mainly speculations about expected results of future work.

Additional comments

• Numerals bellow eleven should be written, not presented by symbol.
• Lines 51-52-It is true that there are fewer studies of photosymbionts in octocorals and sponges but they do date back to 1991 for octocorals (Shick et al. 1991) and early 70s for sponges (Vacelet 1971; Wilkinson and Vacelet 1979, doi:10.1016/0022-0981(79)90028-5). I would suggest changing the sentence appropriately.
• Line 153- this is the only subheading in the results. Either remove it or add more subheadings.

---

## Round 0.2 · Minor Revisions

As indicated, Reviewer 1 is satisfied with the revision, but offers some minor corrections for the final text.

Considering the limited data on sponge abundances on Pacific reefs, please revise the Abstract to something like this beginning line 31:
"Overall sponge abundance and diversity were low, with <1% sponge cover and only 8 putative species identified by... from surveys of 21 sites."

Try to remove phrases like "relatively low" when there is no comparison made and "over 20" if there is a specific number of sites.

Line 110 - "most sites" suggests that there were some sites that were greater than 1%, but your rebuttal letter indicates that cover was well under 1%. It is useful to repeat in your ms. text (as per your rebuttal letter) that you attempted standard survey techniques, but the sponge cover was to low to detect any, and that coral or bare rock were the most common substrata.

Please add GenBank accession numbers.

Reviewer 1 ·

Basic reporting

Line 165 - Change "Blast" to "BLAST." BLAST is an acronym and should be presented in all capital letters.

Line 166 - Change "Genbank" to "GenBank."

Lines 167-171. Morphological validation from Dr. Diaz is valuable; however, I believe the authors' unpublished data on symbiont composition should be removed. It raises many questions (e.g. were all morphotypes screened for symbionts?) that cause distraction and offers relatively little towards the issue at hand (whether morphotypes are the same species).

Experimental design

No comments

Validity of the findings

No comments

Additional comments

The authors have addressed my concerns in this revision. While I had hoped for more morphological descriptions of the studied sponge species, I understand this is beyond their focus. Further, the provision of the additional figure (Fig 2) will aid future work in the area and on these species.

---

## Round 0.3 · accepted · Accept

The authors have addressed all the concerns of the reviewers and editor.